# Content-qualified antenatal care coverage in Lesotho: An ordinal logistic regression analysis of the 2023–2024 demographic and health survey

Kindu Yinges Wondie[1]*, Nuhamin Tesfa Tsega[2], Tazeb Alemu Anteneh[1], Alemneh Tadesse Kassie[1], Berihun Agegn Mengistie[3], Endalk Birrie Wondifraw[4], Getie Mihret Aragaw[3]

1 Department of Clinical Midwifery, School of Midwifery, College of Medicine and Health Sciences, University of Gondar, Gondar, Ethiopia, 2 Department of Women's and Family Health, School of Midwifery, College of Medicine and Health Sciences, University of Gondar, Gondar, Ethiopia, 3 Department of General Midwifery, School of Midwifery, College of Medicine and Health Sciences, University of Gondar, Gondar, Ethiopia, 4 Department of Pediatrics and Child Health Nursing, College of Health Science, Debre Tabor University, Debre Tabor, Ethiopia

* kinduyinges2010@gmail.com

## Abstract

### Background

Despite high antenatal care contact coverage in Lesotho, the content and quality of care remains suboptimal. This study assessed the level and determinants of content-qualified antenatal care coverage using the 2023–2024 Lesotho Demographic and Health Survey.

### Methods

Data were analyzed from 1,112 women aged 15–49 who had a live birth or stillbirth in the 2 years preceding the survey. Content-qualified antenatal care was constructed as a composite score (0–16) incorporating the timing of initiation, number of contacts, attendance of skilled providers, and receipt of 10 key content components. The score was categorized into ordered tertiles (low: 0–12, moderate: 13, high: 14–16). Survey-weighted ordinal logistic regression was used to identify factors associated with receiving higher content-qualified antenatal care.

### Results

Overall, 28.1% of women had low, 32.4% moderate, and 39.5% high content-qualified antenatal care scores. Higher coverage was significantly associated with being married or living with a partner (AOR = 1.78, 95% CI: 1.12–2.83), having secondary (AOR = 2.05, 95% CI: 1.34–3.14) and higher education (AOR = 1.90, 95% CI: 1.21–2.98), using the internet at least once per week (AOR = 1.61, 95% CI: 1.08–2.40), wanted pregnancy (AOR = 2.87, 95% CI: 1.65–5.01), a smaller household

**Data availability statement:** The data used in this study are publicly available from the Demographic and Health Surveys (DHS) Program at https://dhsprogram.com/data/dataset/Lesotho_Standard-DHS_2023.cfm?flag=1. Researchers can request access through the DHS Program website (https://www.dhsprogram.com/data/). After registration and approval of the dataset request, the datasets can be downloaded for research purposes. The authors did not have any special access privileges to the data; all data are available to other researchers under the same conditions. Therefore, the results presented in this study can be fully replicated by obtaining the data from the DHS Program and following the analytical procedures described in the Methods section.

**Funding:** The author(s) received no specific funding for this work.

**Competing interests:** The authors have declared that no competing interests exist.

(AOR = 1.36, 95% CI: 1.05–1.76), and higher community education (AOR = 1.38, 95% CI: 1.09–1.75).

## Conclusion

Although antenatal care contact coverage is high in Lesotho, content-qualified coverage remains low and inequitable. Targeted interventions to improve maternal and community education, internet access, and pregnancy planning are essential to close the quality-coverage gap.

## Introduction

Sub-Saharan Africa (SSA) has the highest global burden of maternal deaths and stillbirths [1,2]. Lesotho has one of the highest maternal mortality ratios globally, with 478 deaths per 100,000 live births [3] and 39 infant deaths per 1000 [4]. Obstetric hemorrhage, hypertensive disorders, and complications of unsafe abortion are the leading causes [5].

Antenatal care (ANC) is defined as skilled healthcare provided to women during pregnancy [6]. ANC is the cornerstone of pregnancy-related complications detection, treatment, and prevention through counseling and advice, maternal and fetal assessments, vaccination, and micronutrient supplementation [7,8].

Lesotho has made strides to improve quality ANC coverage through reforms, including the 2014 National Health Reform, village health worker expansion, and stronger ANC-HIV integration [3,5]. As a result, the percentage of women attending one and four ANC contacts was 93 and 82 respectively. Despite this remarkable achievement in contact coverage, the expected reductions in maternal and neonatal mortality have not been achieved [4]. This discrepancy highlights the critical quality-contact gap, which can widen due to women starting ANC late, not attending the recommended number of contacts, receiving incomplete care content, scarcity of skilled providers, and inaccessible care because of geographic barriers [9–13]. Measuring the current ANCq coverage is a critical part of monitoring and evaluation of Lesotho's move towards achieving quality ANC coverage.

However, traditional ANC coverage metrics cannot measure the actual adequacy and quality of ANC coverage because they rely on the mere number of ANC contacts [14,15]. These binary measures of "adequate" ANC mask quality gradations, exclude women with no ANC contacts, and are restricted to those with at least one visit, limiting their usefulness for tracking inequities and population-level coverage [16,17].

Content-qualified antenatal care (ANCq) addresses this limitation by measuring both service contact (initiation timing and number of visits) and content (receipt of core ANC components from a skilled provider) [17]. In this study, ANCq is a composite score ranging from 0 to 16, where higher scores indicate coverage [16,17]. ANCq has been validated as a reliable indicator of ANC service coverage in low- and middle-income countries [17,18].

Despite Lesotho's health reforms, including the 2014 National Health Reform, village health worker expansion, and stronger ANC-HIV integration, the content and quality of care remains uneven [19,20]. The 2023−24 Lesotho Demographic and Health Survey (LDHS) provides data on indicators of ANC coverage at the national level (timing of initiation, number of contacts, provider skill, and essential services) [4,16,17]. To date, no study has examined ANCq coverage or its determinants in Lesotho using the most recent data. This study addresses this gap by assessing the level of ANCq coverage and identifying factors associated with higher ANCq categories among women who had their last birth within the two years preceding the survey.

## Methods and materials

### Study setting

Lesotho is a small, mountainous, and landlocked country located in Southern Africa, entirely surrounded by South Africa. It covers approximately 30,000 km² and has a population of approximately 2.2 million. It is divided into ten administrative districts, namely Butha-Buthe, Leribe, Berea, Maseru, Mafeteng, Mohale's Hoek, Quthing, Qacha's Nek, Mokhotlong, and Thaba-Tseka, which are further subdivided into constituencies and community councils. Maseru is the capital city and the main urban center [21]. The population is predominantly rural, and the mountainous terrain poses challenges to accessing healthcare services, particularly maternal and newborn care. Lesotho's health system consists of hospitals, health centers, and community health posts. The country has prioritized improving reproductive, maternal, newborn, child, and adolescent health services, including the adoption of the WHO ANC recommendations for a positive pregnancy experience [3].

### Data source and study design

This study is a secondary analysis of data from the 2023−2024 LDHS, a nationally representative cross-sectional survey conducted by the Ministry of Health in collaboration with international partners [4]. Data collection for the 2023−24 LDHS was conducted from November 27, 2023, to February 29, 2024 [4].

The survey followed the DHS-8 methodology, which includes an expanded set of ANC indicators aligned with the updated ANC model of the World Health Organization [4,22]. The pregnancy and postnatal care recode file from the 2023−24 LDHS was used for this analysis [22].

The LDHS employed a two-stage, stratified cluster sampling design using the 2016 Lesotho Population and Housing Census as the sampling frame. Stratification was based on district and type of residence (urban, peri-urban, and rural), resulting in 29 sampling strata, as Butha-Buthe district lacks a peri-urban area. In the first stage, 400 enumeration areas were selected as clusters using probability proportional to size. In the second stage, all households within each selected cluster were listed, and 25 households were systematically chosen through equal probability sampling without replacement. Sampling weights, stratification, and clustering were applied in all analyses to account for this complex survey design. Full survey procedures are detailed in the 2023−24 LDHS final report [4].

### Study population, sample size, and sampling procedure

The base population for this analysis was women aged 15–49 years who had their most recent live birth or stillbirth within two years preceding the survey. Data were collected through face-to-face interviews using standardized, pretested DHS questionnaires adapted to the Lesotho context and administered by trained fieldworkers [4]. Of the 1,811 women aged 15–49 years, 1112 had their most recent live birth or stillbirth within the 2 years before the survey and were included in the final analysis.

### Variable definition and measurement

**Dependent variable.** The outcome variable was ANCq coverage, a composite indicator combining service contact and content dimensions [18]. The ANCq score, ranging from 0 to 16, was calculated by assigning one point for the first-trimester

initiation, two points for at least one skilled provider visit, between one and three points for the number of ANC contacts (one point for 1–3 visits, two points for 4–7 visits, and three points for eight or more visits), and one point each for the receipt of ten content components, namely blood pressure measured, urine sample taken, blood sample taken, fetal heart rate listened to, asked about vaginal bleeding, counselling on maternal diet, counselling on breastfeeding, fundal height measured, at least two doses of tetanus toxoid before birth, and iron-folic acid supplementation [4,17]. The total score was categorized into three ordered levels using weighted tertiles: low (0–12), moderate [13], and high [14–16].

**Independent variables.** Both individual-and community-level variables were examined. Variables were selected based on prior literature and availability in the 2023−2024 LDHS dataset and defined and coded according to the DHS-8 guideline [22]. Community-level variables were generated by aggregating individual responses at the cluster level and dichotomized at the median [23].

Age measures the woman's age in years at the time of the survey and was recorded in 5-year groups [15–49]. It was recoded into 15–24, 25–34, and 35–49 categories.

Current marital status defines the woman's marital status by the time of the survey and was given as never in union, married, living with partner, widowed, divorced, and separated/no longer living together. It was recoded as married or living with a partner versus not married or not in union.

Educational level is the woman's highest level of education by the time of the survey and was recorded as no education, primary, secondary, and higher. It was recoded as no or primary education, secondary, or higher education.

Woman employed defines the woman's response to her employment or working status by the time of the survey and given as yeas and no.

Ownership of a mobile phone was assessed by asking the respondent whether she personally owned a mobile telephone at the time of the survey. Responses were recorded as "yes" or "no."

Ownership of a bank account was determined by asking the respondent whether she had an account in a bank or other financial institution. Responses were recorded as "yes" or "no."

Frequency of internet use measures frequency of woman's internet usage in the past month before the survey. It was categorized as not used at all, less than once a week, at least once a week, and almost every day.

Problem to access health care was defined as reporting at least one of the following barriers to obtaining health services: not wanting to go alone, difficulty obtaining money for treatment, needing permission to seek care, or distance to a health facility. It was coded as "yes" or "no".

History of abortion asks if a woman ever had a terminated pregnancy and coded as "yes" or "no".

Number of children ever born was defined as the total number of live births a woman had and was categorized as 1, 2–4, and ≥ 5 children.

Wanted last pregnancy was assessed by asking the woman whether her most recent birth was wanted at the time she conceived. Responses were recorded as "wanted then," "wanted later," or "wanted no more".

The wealth index is a composite measure of household living standard generated using principal component analysis based on ownership of selected assets and housing characteristics. It was categorized into quintiles: poorest, poorer, middle, richer, and richest.

Household head defines the sex of the head of the household the participant lives and was given as male and female.

Household size was measured as the total number of family members residing in the household at the time of the survey and was recoded into two categories: ≤ 4 and ≥ 5 members.

Type of place of residence characterizes the area of the woman's permanent residence and was coded as urban or rural. District of residence was defined as the geographically distinct administrative district in which the woman permanently resided. It was recorded according to the ten administrative districts of Lesotho [4].

Community-level education was constructed by calculating the proportion of women within each cluster who had attained secondary or higher education. This proportion was dichotomized at the median and categorized as low (below median) and high (at or above median).

Community-level poverty was constructed by calculating the proportion of households within each cluster that fell within the poorest or poorer wealth quintiles. This proportion was dichotomized at the median value and categorized as low (below the median) and high (at or above the median).

Community-level media exposure was generated by calculating the proportion of women within each cluster who reported access to at least one form of media (radio, television, or newspaper). The aggregated proportion was dichotomized at the median value and categorized as low (below the median) and high (at or above the median).

## Data management and statistical analysis

Data cleaning and statistical analysis were conducted using Stata version 17. Prior to analysis, the presence and completeness of the ANCq indicators and all explanatory variables were verified. In accordance with the DHS-8 recode guideline [22], cases with incomplete information on specific components were retained in the denominator where appropriate.

All analyses accounted for the complex DHS sampling design by applying sampling weights, stratification, and primary sampling units.

The distribution of the ANCq score was examined before modeling. The score demonstrated a discrete and highly skewed distribution with clustering at higher values and absence of certain intermediate scores. Given these characteristics, treating ANCq as a continuous outcome was considered inappropriate. A sensitivity analysis modeling ANCq as a continuous variable was conducted to assess robustness.

The ANCq score was therefore categorized into three ordered levels (low, moderate, and high) using weighted tertiles derived from the cumulative weighted distribution. Given the ordinal nature of this outcome, ordered logistic regression was used to examine associations between independent variables and ANCq coverage category [24].

The proportional odds (cumulative logit) model was applied, which estimates the log odds of being in a higher versus lower outcome category while preserving the natural ordering of the response levels [25]. The proportional odds (parallel lines) assumption was assessed using the Brant test [26].

To evaluate potential clustering at the enumeration area level, a multilevel mixed-effects ordered logistic regression model was initially fitted. Model comparison procedures were performed to determine whether a multilevel approach was warranted. Based on these assessments, the final analysis used survey-weighted ordered logistic regression with cluster-robust standard errors.

Ordered logistic regression was preferred over multinomial regression because the outcome categories have a natural order. Unlike multinomial models, which treat categories as nominal and estimate separate parameters for each comparison, the proportional odds model is more parsimonious and statistically efficient, as it estimates a single set of coefficients across cumulative logits [26].

Bivariable regression analysis was performed for both individual- and community-level variables. Variables with a p-value ≤ 0.20 in bivariable analysis were considered candidates for inclusion in the multivariable model. The multivariable model was adjusted for maternal age, educational level, wealth index, parity, type of residence, current marital status, wantedness of the last pregnancy, household size, frequency of internet use, and community-level education and poverty.

In the multivariable ordered logistic regression analysis, adjusted odds ratios (AORs) with 95% confidence intervals (CIs) were reported, and statistical significance was declared at p < 0.05.

## Ethics statement

This study was a secondary analysis of publicly available, anonymized data from the 2023−24 LDHS (The DHS Program - Lesotho: Standard DHS, 2023-24 Dataset). Access to the dataset was granted through the DHS Program's data access portal following a formal request and approval. The protocol for the LDHS received clearance from both the ICF Institutional Review Board and the Lesotho Ministry of Health Research and Ethics Committee [27]. All survey participants provided written informed consent prior to data collection; for minors, consent was obtained from a parent or legal guardian,

and assent was obtained from the minors themselves [27]. As this study involved secondary analysis of de-identified data, no additional ethical approval was required. The research adhered to the DHS Program data usage policy and relevant guidelines for secondary data use (https://dhsprogram.com)

## Results

### Model diagnostics and assumption testing

The distribution of the ANCq score was examined prior to modeling. The score demonstrated a discrete and highly skewed distribution (skewness = −2.73, kurtosis = 12.30; p < 0.001), with clustering at higher values and absence of certain intermediate scores (e.g., no score of 7). Sensitivity analysis modeling ANCq as a continuous variable produced similar directions and levels of statistical significance, supporting the robustness of the categorical approach.

The proportional odds (parallel lines) assumption underlying the ordered logistic regression model was evaluated using the Brant test. The test indicated that the assumption was not violated ($\chi^2 = 0.09$, p = 0.758), supporting the appropriateness of the proportional odds model [26].

To assess clustering at the enumeration area level, a multilevel mixed-effects ordered logistic regression model was fitted. The null model showed a modest but statistically significant intraclass correlation coefficient (ICC = 6.9%, p < 0.001), indicating some degree of clustering. However, after inclusion of individual- and community-level covariates, the ICC decreased to 0.6%, and the likelihood ratio test comparing multilevel and single-level models was not statistically significant (p = 0.390). Therefore, the survey-weighted single-level ordered logistic regression model was retained for the final analysis.

### Content-qualified antenatal care coverage in Lesotho

Overall, 28.2% (n = 313) of women fell into the low ANCq tertile (scores 0–12), 44.3% (n = 493) into the moderate tertile (score 13), and 27.5% (n = 306) into the high tertile (scores 14–16). Combining moderate and high categories, 71.8% of women achieved moderate or high ANCq coverage, indicating that a substantial minority (28.2%) experienced low coverage while the majority received a reasonable level of content-qualified care. The median ANCq score was 14 (interquartile range: 13–15), reflecting a skewed distribution with clustering at higher values.

Regarding service contact dimensions, 51% of women started ANC in the first trimester, 24% attended eight or more visits, while 93.2% received care from a skilled provider at least once. Individual content component coverage was generally high for blood pressure measurement and fetal heart rate assessment (95.2%) and for blood and urine sampling and fundal height measurement (94.9%). Lower coverage was observed for receipt of at least two doses of tetanus toxoid vaccination (59.2%), counseling on maternal diet (76.5%), and being asked about vaginal bleeding (78.2%) (Table 1).

### Background characteristics of the study participants

The median age of the participants was 25 years (interquartile range: 21–31). Most women were married or living with a partner (72.7%), were not currently employed (73.2%), and had attained secondary or higher education (73.3%). More than half (54.7%) reported using the internet almost every day in the past month. The majority of the respondents (61.2%) resided in rural areas, and 67.3% lived in communities with lower poverty levels (Table 2).

### Determinants of content-qualified antenatal care coverage

Several factors were significantly associated with higher ANCq categories in the multivariable ordinal logistic regression analysis (Table 3).

Women who were married or living with a partner had 60% higher odds of being in a higher ANCq category than unmarried or unpartnered women (AOR 1.60, 95% CI: 1.19–2.16). Women with secondary education (AOR 2.05, 95% CI: 1.44–2.92) and higher education (AOR 1.90, 95% CI: 1.13–3.19) had higher odds than those with no or primary education.

**Table 1. Content-qualified antenatal care coverage dimensions in Lesotho (n = 1112).**

| Dimension of ANCq | Category | Frequency | Percentage |
|---|---|---|---|
| Number of ANC contacts attended | 0 | 43 | 3.87 |
| | 1-3 | 172 | 15.47 |
| | 4-7 | 630 | 56.65 |
| | ≥ 8 | 267 | 24.01 |
| ANC initiated in the first trimester | No | 546 | 49.1 |
| | Yes | 566 | 50.9 |
| Skilled provider in at least one ANC contact | No | 76 | 6.83 |
| | Yes | 1,036 | 93.17 |
| Iron-folic acid provided | No | 163 | 14.66 |
| | Yes | 949 | 85.34 |
| Blood sample | No | 57 | 5.13 |
| | Yes | 1,055 | 94.87 |
| Vaccinated with at least two doses of tetanus toxoid before delivery | No | 454 | 40.83 |
| | Yes | 658 | 59.17 |
| Urine sample taken | No | 69 | 6.21 |
| | Yes | 1,043 | 93.79 |
| Fetal heart rate assessed | No | 52 | 4.68 |
| | Yes | 1,060 | 95.32 |
| Blood pressure measured | No | 53 | 4.77 |
| | Yes | 1,059 | 95.23 |
| Fundal height measured | No | 57 | 5.13 |
| | Yes | 1,055 | 94.87 |
| Asked about vaginal bleeding | No | 243 | 21.85 |
| | Yes | 869 | 78.15 |
| Counseled about diet | No | 261 | 23.47 |
| | Yes | 851 | 76.53 |
| Counseled about breastfeeding | No | 160 | 14.39 |
| | Yes | 952 | 85.61 |

Women who used the internet at least once per week had 61% higher odds than those who did not use the internet (AOR 1.61, 95% CI: 1.02–2.56). Women whose last pregnancy was wanted at the time of conception had nearly three times higher odds than those whose pregnancy was not wanted (AOR 2.87, 95% CI: 2.04–4.05).

Smaller household size (≤ 4 members) was associated with 36% higher odds (AOR 1.36, 95% CI: 1.04–1.78). At the community level, residence in clusters with a higher proportion of women who had completed secondary or higher education was associated with 38% higher odds (AOR, 1.38; 95% CI: 1.02–1.81) (Table 3).

## Discussion

This study provides the first assessment of ANCq coverage and its determinants in Lesotho using the 2023–2024 DHS. Overall, 71.9% of women achieved moderate or high ANCq scores, with a median score of 14, indicating that the full package of recommended services is not universally received although ANC attendance is high.

The magnitude of ANCq coverage in Lesotho is relatively higher than that of earlier multi-country analyses from SSA and South Asia, which reported much lower proportions using narrower content indicators or stricter binary definitions of adequacy. For example, previous studies reported low adequate ANC coverage in SSA (3%) [23], Rwanda (8.9%) [28],

**Table 2.  Distribution of receipt of content-qualified antenatal care by background characteristics of women in Lesotho (N = 1112).**

| Variable | Category | Content qualified antenatal care coverage | | | Total (%) | P-value |
|---|---|---|---|---|---|---|
| | | Low | Moderate | High | | |
| Age | 15-24 | 151 | 235 | 138 | 524 (43.3) | 0.952 |
| | 25-34 | 113 | 189 | 128 | 430 (41.4) | |
| | 35-49 | 49 | 69 | 40 | 158 (15.3) | |
| Current marital status | Married | 201 | 335 | 243 | 779 (72.7) | 0.000 |
| | Unmarried | 112 | 158 | 63 | 333 (27.3) | |
| Household head | Male | 197 | 286 | 198 | 681 (64.5) | 0.061 |
| | Female | 116 | 207 | 108 | 431 (35.5) | |
| Educational level | Up to primary | 131 | 163 | 62 | 356 (26.8) | 0.000 |
| | Secondary | 163 | 285 | 198 | 646 (58.7) | |
| | Higher | 19 | 45 | 46 | 110 (14.6) | |
| Currently employed | No | 252 | 406 | 224 | 882 (73.2) | 0.009 |
| | Yes | 61 | 87 | 82 | 230 (26.8) | |
| Wealth index | Poorest | 124 | 173 | 73 | 370 (21.7) | 0.000 |
| | Poorer | 68 | 98 | 46 | 212 (17.2) | |
| | Middle | 57 | 97 | 65 | 219 (21.8) | |
| | Richer | 38 | 80 | 60 | 178 (20.5) | |
| | Richest | 26 | 45 | 62 | 133 (18.8) | |
| Owns mobile phone | No | 83 | 104 | 41 | 228 (16.7) | 0.002 |
| | Yes | 230 | 389 | 265 | 884 (83.3) | |
| Owns bank account | No | 259 | 374 | 205 | 838 (68.7) | 0.001 |
| | Yes | 54 | 119 | 101 | 274 (31.3) | |
| Frequency of internet use | Not at all | 121 | 160 | 68 | 349 (26.5) | 0.000 |
| | Less than once a week | 30 | 43 | 13 | 86 (8.0) | |
| | At least once a week | 32 | 70 | 40 | 142 (10.8) | |
| | Almost every day | 130 | 220 | 185 | 535 (54.7) | |
| Problem to access health care | No | 160 | 283 | 190 | 633 (64.3) | 0.165 |
| | Yes | 153 | 210 | 116 | 479 (35.7) | |
| History of abortion | No | 269 | 445 | 264 | 978 (87.6) | 0.071 |
| | Yes | 44 | 48 | 42 | 134 (12.4) | |
| Number of children ever born | 1 | 115 | 229 | 147 | 491 (45.1) | 0.195 |
| | 2-4 | 168 | 222 | 145 | 535 (48.5) | |
| | ≥ 5 | 27 | 39 | 10 | 76 (6.4) | |
| Family size | ≤ 4 | 88 | 180 | 125 | 393 (38.6) | 0.000 |
| | ≥ 5 | 225 | 313 | 181 | 719 (61.4) | |
| Last pregnancy wanted | Then | 91 | 201 | 181 | 473 (45.7) | 0.000 |
| | Later | 117 | 168 | 84 | 369 (31.5) | |
| | No more | 105 | 124 | 41 | 270 (22.8) | |
| Type of place of residence | Urban | 70 | 140 | 114 | 324 (38.8) | 0.010 |
| | Rural | 243 | 353 | 192 | 788 (61.2) | |
| District of residence | Butha-Buthe | 31 | 45 | 44 | 120 (10.8) | 0.077 |
| | Leribe | 34 | 49 | 37 | 120 (10.8) | |
| | Berea | 34 | 46 | 32 | 112 (10.1) | |
| | Maseru | 28 | 55 | 44 | 127 (11.4) | |
| | Mafeteng | 27 | 38 | 15 | 80 (7.2) | |

*(Continued)*

**Table 2.** (Continued)

| Variable | Category | Content qualified antenatal care coverage | | | Total (%) | P-value |
|---|---|---|---|---|---|---|
| | | Low | Moderate | High | | |
| | Mohale's Hoek | 33 | 55 | 23 | 111 (10.0) | |
| | Quthing | 27 | 35 | 21 | 83 (7.5) | |
| | Qacha's Nek | 21 | 45 | 28 | 94 (8.5) | |
| | Mokhotlong | 32 | 60 | 27 | 119 (10.7) | |
| | Thaba-Tseka | 46 | 65 | 35 | 146 (13.1) | |
| Community-level education | Low | 114 | 171 | 82 | 367(27.9) | 0.133 |
| | High | 199 | 322 | 224 | 745(72.1) | |
| Community-level poverty | Low | 146 | 255 | 197 | 598(67.3) | 0.000 |
| | High | 167 | 238 | 109 | 514(32.7) | |
| Community-level media exposure | Low | 217 | 328 | 165 | 710(51.9) | 0.001 |
| | High | 96 | 165 | 141 | 402(48.1) | |

P-values from design-based Pearson χ² test.

Tanzania (43%) [29], India (13–32%) [30–32], and Bangladesh (18%) [33]. These discrepancies can be explained by differences in measurement approaches, survey instruments, and contextual factors.

ANCq coverage was measured using an ordered scoring system that captures variation across low, medium, and high levels rather than collapsing women into only "adequate" versus "inadequate" categories. In addition, the DHS-8 questionnaire includes a broader set of ANC content indicators and applies a shorter recall period than earlier DHS phases, which may improve reporting accuracy and increase observed ANCq coverage. Differences in the number, type, and accessibility of ANC services across countries also contribute to variation in ANCq coverage. For instance, this analysis included fundal height measurement, and Lesotho's integrated HIV-ANC programs may have strengthened the delivery of comprehensive services compared with other SSA settings.

In contrast, many previous studies [28–33] defined ANCq coverage using strict binary (adequate vs. inadequate) criteria requiring fulfilment of all recommended contacts and services for adequate ANCq coverage. This binary definition also condenses the scores to two extremes.

Several factors, including current marital status, maternal education level, frequency of internet use, household size, wantedness of the last pregnancy, and community-level education, were significantly associated with ANCq coverage.

Marital status was significantly associated with women's receiving of ANCq, with married or living in union having 60% higher odds of being in a higher ANCq category compared with unmarried women. This finding is consistent with evidence from the SSA [34,35]. The association likely reflects the economic and social support from partners. Financial security within marriage facilitates allocation of resources to ANC, including transport, user fees, and compensation for lost income during clinic visits, while spousal involvement can enhance women's autonomy, provide emotional support, and promote completion of recommended contacts. In Lesotho, where family and spousal structures strongly influence health-seeking behavior, these effects are amplified. Moreover, stigmatization of pregnancy outside marriage is Lesotho could discourage unmarried women from engaging with ANC services [36], whereas women in unions are more likely to receive encouragement and practical assistance from partners and extended family networks [37]. Evidence from culturally similar Sesotho-speaking communities shows that male involvement reduces maternal stress, improves couple communication, and promotes shared responsibility for health decisions, thereby strengthening ANC content and effectiveness [38–41].

 

**Table 3. Bi-variable and multivariable ordinal logistic regression analysis of factors associated with receiving content-qualified antenatal care in Lesotho 2023–24.**

| Variable | Category | COR (95% CI) | AOR (95% CI) |
|---|---|---|---|
| Current marital Status | Unmarried/not living with partner | 1 | 1 |
| | Married/living with partner | 2.10(1.59-2.76) | 1.60(1.19-2.16) ** |
| Education level | Up to primary | 1 | 1 |
| | Secondary | 2.19(1.64-2.91) | 2.05(1.44-2.92) *** |
| | Higher | 3.19(2.13-4.79) | 1.90(1.13-3.19) * |
| Currently employed | No | 1 | 1 |
| | Yes | 1.59(1.20-2.12) | 1.08(0.78-1.48) |
| Wealth Index | Poorest | 1 | 1 |
| | Poorer | 1.06(0.72-1.55) | 1.00(0.66-1.51) |
| | Middle | 2.02(1.40-2.92) | 1.56(0.94-2.60) |
| | Richer | 2.39(1.64-3.49) | 1.60(0.90-2.84) |
| | Richest | 3.25(2.18-4.83) | 1.67(0.89-3.10) |
| Mobile phone | No | 1 | 1 |
| | Yes | 1.88(1.35-2.61) | 0.99(0.67-1.45) |
| Owns a bank account | No | 1 | 1 |
| | Yes | 1.86(1.43-2.43) | 1.23(0.90-1.67) |
| Frequency of internet use in the past month | Never | 1 | 1 |
| | Less than once a week | 0.71(0.44-1.14) | 0.70(0.42-1.16) |
| | At least once a week | 1.83(1.19-2.81) | 1.61(1.02-2.56) * |
| | Almost every day | 2.12(1.59-2.83) | 1.31(0.91-1.87) |
| Problem to access health care | No | 1.33(1.03-1.71) | 0.94(0.71-1.24) |
| | Yes | 1 | 1 |
| Parity | 1 | 2.00(1.21-3.30) | 0.90(0.5-1.58) |
| | 2-4 | 1.62(0.98-2.68) | 0.66(0.38-1.14) |
| | ≥ 5 | 1 | 1 |
| Household size | 1-4 | 1.76(1.37-2.27) | 1.36(1.04-1.78) * |
| | ≥ 5 | 1 | 1 |
| Last pregnancy wanted | Then | 3.86(2.81-5.31) | 2.87(2.04-4.05) *** |
| | Later | 1.47(1.06-2.03) | 1.29(0.91-1.82) |
| | No more | 1 | 1 |
| Type of place of residence | Rural | 1 | 1 |
| | Urban | 1.73(1.32-2.28) | 0.95(0.66-1.35) |
| Community-level education | Low | 1 | 1 |
| | High | 1.39(1.04-1.87) | 1.38(1.02-1.81) *** |
| Community-level poverty | Low | 1.89(1.44-2.47) | 1.88(1.42-2.10) |
| | High | 1 | 1 |
| Community-level media exposure | Low | 1 | 1 |
| | High | 1.76(1.35-2.30) | 1.37(0.98-1.91) |

AOR = adjusted odds ratio; COR = crude odds ratio; CI = confidence interval.

*, p < 0.05; **, p < 0.01; ***, p < 0.001.

Male partner involvement is also associated with earlier initiation, higher attendance, and better adherence to key interventions such as blood testing, contributing to improved overall ANC quality through consistent follow-up and compliance with care protocols [42].

Maternal education is a key determinant of ANCq. Women who had attained secondary or higher education had nearly twice the odds of being in a higher ANCq category than those with primary education or none. This finding is consistent with those of previous DHS analyses from Lesotho and across SSA [23,43]. The association likely reflects differences in knowledge and health literacy: more educated women are better informed about the benefits of early booking and adherence to the recommended number and content of ANC contacts, and therefore place greater value on these services. Higher educational attainment also enhances women's ability to recognize danger signs of pregnancy and understand the consequences of unqualified care [44]. In the Lesotho context, education further improves women's awareness of recommended ANC components and their capacity to effectively navigate the health system, reinforcing receipt of higher ANCq [45].

Frequency of internet use has emerged as an important determinant of ANCq coverage. Women who reported using the internet at least once per week had 60% higher odds of being in a higher ANCq category than those who reported no use. A recent Tanzanian DHS analysis found that women who reported using internet had lower dropping out from the maternal continuum of care compared to those with no internet exposure [46]. Regular internet use may expose women to maternal health information that improves their understanding of recommended services and danger signs [47]. In Lesotho, where misconceptions and cultural beliefs can delay ANC care-seeking [36], access to reliable online information may help counter misinformation and support appropriate use of ANC services [37].

Intention to last pregnancy by the time of conception was also strongly associated with ANCq. Women whose last pregnancy was wanted at the time had nearly three times the odds of being in a higher ANCq category compared with pregnancies unwanted at the time of conception. This finding is consistent with DHS analyses from Rwanda, Somalia, and Bangladesh, as well as systematic reviews from Ethiopia [28,48,49]. Unintended pregnancies are often accompanied by emotional stress, reduced motivation, and delayed engagement with health services. These factors can lead to later initiation of ANC, fewer visits, and reduced uptake of recommended service components, ultimately lowering ANC quality. Unintended pregnancies are often accompanied by emotional stress, reduced motivation, and delayed engagement with health services [48]. These factors can lead to later ANC initiation, fewer visits, and reduced uptake of recommended service components, ultimately lowering ANC quality [43].

Household size showed an inverse relationship with the ANCq coverage. Women living in smaller households had higher odds of being in a higher ANCq category than those living in larger households. This finding is supported by a DHS analysis report in SSA [23]. Women in larger households in extended family structures common in Lesotho may face competing domestic and caregiving responsibilities, reduced autonomy, and time constraints that limit repeated ANC attendance [50]. Women with several children may also rely on prior pregnancy experience or avoid facilities because of fear of criticism, reducing their likelihood of receiving complete ANC content [36].

Higher community-level education was significantly associated with greater ANCq coverage. Women residing in communities with higher levels of education had approximately 40% greater odds of being in a higher ANCq category compared with those from less educated communities. This finding is consistent with multilevel analyses from sub-Saharan Africa showing that community education strongly influences the quality and adequacy of antenatal care received by women [51,52]. The association likely reflects the broader social and cultural environment shaped by community-level education, which promotes maternal healthcare norms. Women are more likely to utilize antenatal care services adequately in communities with higher female literacy and more gender-equitable norms, independent of their individual socioeconomic status.

## Conclusion

This study highlights that while most women in Lesotho attend antenatal care, many do not receive the full package of recommended services. ANCq coverage is strongly shaped by socioeconomic position and information access rather than by simple service contact alone. Marital status, maternal education, internet use frequency, pregnancy intention,

household size, and community-level education were found to be significant determinants of being in a higher ANCq coverage category.

These findings indicate that, beyond contact coverage, Lesotho needs to emphasize on provision of quality and equitable ANC. Strengthening educational opportunities for women, expanding access to the internet, and promoting planned pregnancy are essential steps toward ensuring that all pregnant women receive content-qualified antenatal care.

## Strength and limitations

This study measured ANCq coverage at the population level and included a wider range of ANC contents. This helps to better understand Lesotho's progress in ensuring quality ANC. This study employed an ordinal logistic regression analysis, preserving the natural ordering inherent in the outcome variable, preventing arbitrary dichotomization of ordinal data, which would have resulted in information loss. Research demonstrates that "for categorical data with ordered categories, ordinal logistic regression works well" and provides advantages over other models [24].

The study used a self-reported data, which may be affected by recall or social desirability bias, although the shorter DHS-8 recall period likely minimized it. The ANC content indicators available in DHS-8 do not encompass all components recommended by the WHO 2016 ANC mode; therefore, the ANCq measure reflects only the interventions captured in the survey. Moreover, the dataset does not provide information on the timing or sequencing of individual ANC components, limiting the assessment of whether services were delivered at clinically appropriate points during pregnancy. Important facility- and system-level determinants, such as provider adherence to ANC protocols and service readiness, were not captured, restricting the examination of ANCq coverage's structural drivers.

## Author contributions

**Conceptualization:** Kindu Yinges Wondie.

**Data curation:** Kindu Yinges Wondie.

**Formal analysis:** Kindu Yinges Wondie, Nuhamin Tesfa Tsega, Tazeb Alemu Anteneh, Alemneh Tadesse Kassie, Berihun Agegn Mengistie, Endalk Birrie Wondifraw, Getie Mihret Aragaw.

**Methodology:** Kindu Yinges Wondie, Getie Mihret Aragaw.

**Writing – original draft:** Kindu Yinges Wondie.

**Writing – review & editing:** Kindu Yinges Wondie, Nuhamin Tesfa Tsega, Tazeb Alemu Anteneh, Alemneh Tadesse Kassie, Berihun Agegn Mengistie, Endalk Birrie Wondifraw, Getie Mihret Aragaw.

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
