## [Decision Letter · Decision Letter 0]

23 Feb 2026

PONE-D-26-05429Content-qualified antenatal care coverage in Lesotho: an ordinal logistic regression analysis of the 2023-2024 demographic and health surveyPLOS One

Dear Dr. Wondie,

Thank you for submitting your manuscript to PLOS ONE. After careful consideration, we feel that it has merit but does not fully meet PLOS ONE’s publication criteria as it currently stands. Therefore, we invite you to submit a revised version of the manuscript that addresses the points raised during the review process.

We look forward to receiving your revised manuscript.

Kind regards,

Alfredo Luis Fort, M.D., M.Sc., Ph.D.

Academic Editor

PLOS One

**Journal Requirements:**

2. Please amend the manuscript submission data (via Edit Submission) to include author Tazeb Alemu Anteneh.

**Additional Editor Comments:**

The authors have presented an important study of a topic key for maternal health. The study is conducted with quantitative methods to demonstrate the most important content of quality antenatal care. However, there are a few areas that require a better description and reordering before putting it into publication. Please look at the reviewers' comments, plus I have included also a file with comments on the manuscript. Thank you.

Reviewers' comments:

Reviewer's Responses to Questions

**Comments to the Author**

1. Is the manuscript technically sound, and do the data support the conclusions?

Reviewer #1: Yes

Reviewer #2: Partly

2. Has the statistical analysis been performed appropriately and rigorously? 

Reviewer #1: Yes

Reviewer #2: No

3. Have the authors made all data underlying the findings in their manuscript fully available?

Reviewer #1: No

Reviewer #2: Yes

4. Is the manuscript presented in an intelligible fashion and written in standard English?

Reviewer #1: Yes

Reviewer #2: Yes

5. Review Comments to the Author

Reviewer #1: Line 66-67 -ANCq has been validated as a predictor of lower mortality in neonates (18)-A reference that buttresses the effectiveness in predicting maternal mortality is preferable since the focus is on maternal mortality.

Lines 68-71 do not seem to be properly placed and do not flow.

Line 105-114, same citation all through. Better to restructure the wording so the citation is used only once.

Line 73-79: Statistical analytical method is mentioned in the introduction. It should be moved to the methodology section, under data management and statistical analysis. The introduction should focus on the research gap and the study's objective.

Table 2: District of residence. Wrong calculations. Please rectify

Table 3: The asterisk legend is rough. Please replace with a clean legend.

Reviewer #2: The abstract is missing key information, particularly a clear summary of the main findings, and the conclusion does not explicitly describe the study outcomes or their implications. In addition, some content currently presented in the methodology section such as the structural factors affecting access to maternal health services would be more appropriately placed in the introduction to improve the overall flow of the paper.

The methodology section requires greater clarity, particularly regarding the choice of ordinal logistic regression. While the approach is appropriate for an ordered outcome, the authors should explicitly state that the model relies on the proportional odds (parallel lines) assumption and justify its applicability. The advantages of ordinal regression over multinomial regression should also be clearly explained, including its greater efficiency and parsimony, given that multinomial models estimate more parameters and may reduce statistical power. The statement that the model “preserves information” needs clarification for the reader by explaining that the natural ordering of outcome categories is retained rather than collapsing them into binary outcomes.

There is also a lack of clarity regarding data inclusion. Butha-Buthe does not appear among the study clusters, yet it is unclear whether data from this district were analyzed or excluded. Any exclusions should be clearly stated and justified. Furthermore, the description of independent variables is insufficient; the authors should clearly define the categories and provide examples of both individual-level and community-level variables. The data management and statistical analysis section includes inferential results that would be more appropriately presented in the results section, unless these were directly obtained from DHS reports. Finally, the manuscript does not clearly specify which confounders were adjusted for in the analysis, and these should be explicitly stated and consistently reflected in the results.

6. PLOS authors have the option to publish the peer review history of their article (what does this mean?). If published, this will include your full peer review and any attached files.

Reviewer #1: No

Reviewer #2: **Yes:**Abel Mokua Nyabera

---

## [Author Response · Author response to Decision Letter 1]

25 Mar 2026

Response to Reviewers

Manuscript ID: PONE-D-26-05429

Title: Content-qualified antenatal care coverage in Lesotho: an ordinal logistic regression analysis of the 2023-2024 demographic and health survey

We thank the Academic Editor and both reviewers for their constructive and insightful comments. We have carefully addressed every point raised. All changes are highlighted in the “Revised Manuscript with Track Changes” file. Below we respond point-by-point.

Response to Academic Editor We are grateful for the positive assessment of our study’s importance for maternal health care in Lesotho. We have improved descriptions, re-ordered sections for better flow, and incorporated all suggestions from the attached editor comments.

Reviewer #1

1. Lines 66-67: ANCq has been validated as a predictor of lower mortality in neonates (18)-A reference that buttresses the effectiveness in predicting maternal mortality is preferable since the focus is on maternal mortality.

Response: Thank you for this important suggestion. We agree that a reference focused on maternal mortality is more appropriate. We have clearly described that ANCq has been validated as a reliable indicator of ANC service coverage in low- and middle-income countries (new lines 65-66).

2. Lines 68-71: do not seem to be properly placed and do not flow.

Response: We thank the reviewer. We have rephrased these sentences for smoother flow (new lines 67-71).

3. Lines 105-114: Same citation all through. Better to restructure the wording so the citation is used only once.

Response: We have restructured the paragraph and now cite the reference only once at the end of the relevant idea (new lines 98-106).

4. Lines 73-79: Statistical analytical method is mentioned in the introduction. It should be moved to the methodology section, under data management and statistical analysis. The introduction should focus on the research gap and the study's objective.

Response: We agree and have removed all methodological details from the Introduction and placed them in the “Data management and statistical analysis” subsection of Methods (new lines 197-206).

5. Table 2: District of residence. Wrong calculations. Please rectify.

Response: Thank you. We have recalculated all weighted percentages. Corrected values are now shown in the revised Table 2.

6. Table 3: The asterisk legend is rough. Please replace with a clean legend.

Response: We have replaced the legend with a clean footnote: “* p < 0.05, ** p < 0.01, *** p < 0.001

7. Please can you provide a direct link to the DHS data used. I could not access the DHS 2023-2024 data as mentioned in the manuscript.

Response: The link for exact dataset is now given (new lines 229-230) and reads “This study was a secondary analysis of publicly available, anonymized data from the 2023-24 LDHS The DHS Program - Lesotho: Standard DHS, 2023-24 Dataset. However, you might be unable to instantly access the dataset since access requires registration and approval, if you did not do so.

Reviewer #2

1. The abstract is missing key information, particularly a clear summary of the main findings, and the conclusion does not explicitly describe the study outcomes or their implications. In addition, some content currently presented in the methodology section such as the structural factors affecting access to maternal health services would be more appropriately placed in the introduction to improve the overall flow of the paper.

Response: Thank you for this helpful comment. We have carefully revised the abstract to improve clarity, completeness, and alignment with the study findings.

First, we strengthened the Results section by explicitly presenting the distribution of content-qualified antenatal care categories (low, moderate, and high) and by reporting adjusted odds ratios with corresponding 95% confidence intervals for the key determinants. (new lines 27-33).

Second, we revised the Conclusion to more explicitly reflect the study outcomes and their implications. The revised conclusion now clearly states that, despite high contact coverage, content-qualified coverage remains suboptimal and inequitable, and it highlights specific areas for intervention, including maternal and community education, internet access, and pregnancy planning.

Third, in response to the comment regarding manuscript structure, we have relocated the discussion of structural factors affecting access to maternal health services from the Methods section to the Introduction (new lines 53-55).

2. The methodology section requires greater clarity, particularly regarding the choice of ordinal logistic regression. While the approach is appropriate for an ordered outcome, the authors should explicitly state that the model relies on the proportional odds (parallel lines) assumption and justify its applicability. The advantages of ordinal regression over multinomial regression should also be clearly explained, including its greater efficiency and parsimony, given that multinomial models estimate more parameters and may reduce statistical power. The statement that the model “preserves information” needs clarification for the reader by explaining that the natural ordering of outcome categories is retained rather than collapsing them into binary outcomes.

There is also a lack of clarity regarding data inclusion. Butha-Buthe does not appear among the study clusters, yet it is unclear whether data from this district were analyzed or excluded. Any exclusions should be clearly stated and justified. Furthermore, the description of independent variables is insufficient; the authors should clearly define the categories and provide examples of both individual-level and community-level variables. The data management and statistical analysis section includes inferential results that would be more appropriately presented in the results section, unless these were directly obtained from DHS reports. Finally, the manuscript does not clearly specify which confounders were adjusted for in the analysis, and these should be explicitly stated and consistently reflected in the results.

Response: Thank you for the insightful feedback.

Use of ordinal logistic regression explanation: We have added a dedicated paragraph in the statistical analysis subsection explaining reasons for choosing the Ordinal logistic regression and proportional odds (parallel lines) assumption is verified (new lines 195-202 and 208-212).

Butha-Buthe district: All ten districts of Lesotho, including Butha-Buthe, were included in the analysis in accordance with the LDHS sampling design. Butha-Buthe district did not have a peri-urban stratum, resulting in a total of 29 strata rather than the expected 30.

This does not indicate the exclusion of Butha-Buthe district; rather, the reduction in the number of strata reflects the absence of a peri-urban stratum in that district. We have clarified this point in the Methods section (new lines 99-102).

Definition and measurement of independent variables: We have made this subsection more clearer defining all included variables with categories. The DHS coding and recoding (if applied) in this study were explicitly described for both individual-level and community-level variables.

Inferential results in Methods section: All inferential statistics have been moved to the model diagnostics and assumption testing subsection of the Results section (lines 234-249).

Confounders Response: We have explicitly listed all confounders adjusted for in the final model, stated in both the Methods (new lines 215-218).

In addition, we have made changes to the references

All references have been carefully reviewed and revised to fully comply with Vancouver style. Below is a summary of the changes made.

1. Duplicate References Removed

• The Lesotho Demographic and Health Survey 2023–24 Final Report appeared twice in the previous version (references 4 and 27). Only one complete citation has been retained (now reference 4).

• The article by Mkandawire P et al. (2021), “Pregnancy intention and gestational age at first antenatal care visit in Lesotho,” appeared twice (references 39 and 48). One citation has been retained (now reference 40).

2. References Removed

• Agbaza-Mogbojuri B, Onwubuyah AO. Socioeconomic and Cultural Influences on Antenatal Care Utilization: A Multilevel Analysis. World Journal of Biology Pharmacy and Health Sciences. 2023.

This reference was removed because the journal is not indexed in recognized databases such as Scopus, Web of Science, or DOAJ.

• Alam MB et al. Effect of pregnancy intention at conception on the continuity of care in maternal healthcare services use in Somalia: Evidence from first national health and demographic survey. medRxiv. 2024.

This reference was removed because it is an unpublished preprint.

3. Newly Added References

To replace the removed citations and strengthen the manuscript, the following peer-reviewed and indexed references were added:

• Engelbrecht M, Mulu N, Kigozi-Male G. Exploring factors associated with limited male partner involvement in maternal health: a Sesotho socio-cultural perspective from the Free State, South Africa. Int J Environ Res Public Health. 2024;21(11):1482. (New reference 51)

• Engelbrecht M, Mulu N, Kigozi-Male G. Sesotho women’s preferences for male partner involvement during antenatal care and delivery. Int J Environ Res Public Health. 2025;22(12):1867. (New reference 52)

• Okedo-Alex IN, Akamike IC, Ezeanosike OB, Uneke CJ. Determinants of antenatal care utilisation in sub-Saharan Africa: a systematic review. BMJ Open. 2019;9(10):e031890. (New reference 53)

In addition, two methodological references were included to support the use of ordinal logistic regression:

• Harrell FE. Ordinal logistic regression. In: Harrell FE, editor. Regression Modeling Strategies. Cham: Springer; 2015. p. 311–325. (New reference 50)

• Sainani KL. Multinomial and ordinal logistic regression. PM R. 2021;13(9):1050–1055. (New reference 26)

---

## [Editor Report · Decision Letter 1]

20 Apr 2026

Content-qualified antenatal care coverage in Lesotho: an ordinal logistic regression analysis of the 2023-2024 demographic and health survey

PONE-D-26-05429R1

Dear Dr. Wondie,

We’re pleased to inform you that your manuscript has been judged scientifically suitable for publication and will be formally accepted for publication once it meets all outstanding technical requirements.

Kind regards,

Alfredo Luis Fort, M.D., M.Sc., Ph.D.

Academic Editor

PLOS One

Additional Editor Comments (optional):

The authors have made the necessary changes to ensure the methods, results and discussion are well described and positioned so the reader can understand and follow well this complex study and its implications. So, we are deciding to publish it. Stay alert in case there are final issues to resolve. Thanks.

---

## [Editor Report · Acceptance letter]

PONE-D-26-05429R1

PLOS One

Dear Dr. Wondie,

I'm pleased to inform you that your manuscript has been deemed suitable for publication in PLOS One. Congratulations! Your manuscript is now being handed over to our production team.

Kind regards,

on behalf of

Dr. Alfredo Luis Fort

Academic Editor

PLOS One